# Creating Healthy Change in the Preconception Period for Women with Overweight or Obesity: A Qualitative Study Using the Information–Motivation–Behavioural Skills Model

**DOI:** 10.3390/jcm9103351

**Published:** 2020-10-19

**Authors:** Jodie Scott, Melissa Oxlad, Jodie Dodd, Claudia Szabo, Andrea Deussen, Deborah Turnbull

**Affiliations:** 1School of Psychology, The University of Adelaide, Adelaide 5005, Australia; melissa.oxlad@adelaide.edu.au (M.O.); deborah.turnbull@adelaide.edu.au (D.T.); 2Robinson Institute and School of Paediatrics and Reproductive Health, Department of Perinatal Medicine, Women’s and Babies Division, Women’s & Children’s Hospital, The University of Adelaide, Adelaide 5005, Australia; jodie.dodd@adelaide.edu.au (J.D.); andrea.deussen@adelaide.edu.au (A.D.); 3School of Computer Science, The University of Adelaide, Adelaide 5005, Australia; claudia.szabo@adelaide.edu.au

**Keywords:** preconception, weight management, overweight, obesity, behaviour change, healthy lifestyle, maternal health

## Abstract

Worldwide, half of women begin a pregnancy with overweight or obesity, which increases the risk of pregnancy and birth complications and adversely affects the lifelong health of the offspring. In order for metabolic changes to influence the gestational environment, research suggests that weight loss should take place before conception. This study aimed to understand women’s emotional and social contexts, knowledge, motivations, skills and self-efficacy in making healthy change. Semi-structured interviews conducted with twenty-three women with overweight or obesity, informed by the Information–Motivation–Behavioural Skills (IMB) model, were analysed using reflexive thematic analysis. Information-related themes identified included poor health risk knowledge, healthy food decisions and health versus convenience. The Motivation themes comprised taking responsibility, flexible options, social expectations, interpersonal challenges and accountability. Behavioural Skills entailed themes such as the mental battle, time management, self-care and inspiration. An environmental factor was identified in affordability—limiting access to healthier alternatives. Women wanted simple, flexible options that considered family commitments, time and budgetary constraints. Unprompted, several mentioned the importance of psychological support in managing setbacks, stress and maintaining motivation. Strategies for enhancing self-efficacy and motivational support are required to enable longstanding health behaviour change. Findings will inform intervention mapping development of an eHealth solution for women preconception.

## 1. Introduction

Worldwide, half of all women of childbearing age have overweight (body mass index (BMI) ≥ 25.0 to 29.9 kg/m^2^) or obesity (BMI ≥ 30 kg/m^2^) [1]. This figure is significant as retrospective, case-control and cohort studies have found that women who enter pregnancy with obesity are at higher risk of gestational diabetes mellitus (GDM) [2] and pre-eclampsia [2,3]—both associated with long-term morbidities [4]. A review of reviews on the risks of maternal obesity also found these women are more likely to experience early pregnancy loss, premature birth and stillbirth, have a higher risk of foetal malformations [4] and are less likely to initiate [2,5] and maintain breastfeeding [2,5]. The infant is also more likely to be born with a high birth weight (macrosomia) [2], which can result in an increased risk of birth trauma and maternal complications [6] including the need for induction or caesarean birth [2], and higher risk of postpartum haemorrhage [7].

It is clear that the gestational environment also plays a critical role in the long-term health of the offspring. Birth cohort longitudinal studies show that maternal obesity is the strongest risk factor for offspring obesity across childhood, adolescence and adulthood [8]. Both epidemiological studies and animal models suggest that infants born with a high birthweight are also predisposed to metabolic syndrome, which, in turn, increases the risk of heart disease, stroke and type 2 diabetes mellitus in later life [9]. Analyses of prospective, observational and longitudinal studies also support an association between maternal obesity and both poorer cognitive function and mental health in the offspring across the lifespan, via epigenetic mechanisms [10]. These risks represent a significant public health problem and the resulting costs to both the health system, and society, are considerable [9].

### 1.1. Targeting the Preconception Period

Randomised control trials and meta-analyses have found that, while dietary and exercise intervention strategies initiated during pregnancy reduce gestational weight gain and lower the risk of caesarean section, they are insufficient to impact other clinical outcomes [1,11] for either mother or baby. Significantly, no effect has been seen in offspring outcomes [1]. In order for the metabolic effects of weight loss to take place, and for the gestational environment to be exposed to these changes, it is recommended that weight management take place before conception [12,13,14]. Biological parenting begins even before conception, as the female gametes provide the embryo with a genomic blueprint at the point of fertilisation [15]. Preconception obesity can change the phenotype and potential of gametes and early embryos [16] as transgenerational epigenetic markers interact with the conditions at conception, effectively programming the developmental path of the embryo [15]. The period before conception is therefore a critical window for optimising developmental programming to reduce the incidence of metabolic, cardiovascular, immune, and neurological morbidities in the offspring across the lifespan [16].

While public health guidelines note that BMI before pregnancy is independently associated with pregnancy outcomes [17], there is currently no preconception healthcare program routinely used within Australia specifically aimed at helping women manage their weight before pregnancy [18]. The World Health Organization highlighted preconception pregnancy care as one of six key priorities for ending childhood obesity [19,20]. The perinatal research community has expressed the urgent need for evaluation of preconception public health strategies with regard to obesity [21]. Guidelines for obesity management in women planning a pregnancy are currently based on a consensus view and lack evidence supporting efficacy [13,22].

Nutrition and physical exercise are potentially highly modifiable factors with underestimated potential for improvements in both the short and long-term health of mother and baby [23]. While many studies have explored weight maintenance in pregnant women, focused primarily on limiting gestational weight gain, few have targeted the vital preconception phase [12,24]. Women who intend to become pregnant are more amenable to lifestyle advice, so interventions during this time may be more effective than during any other period in life [25].

### 1.2. Weight Management Interventions

Traditional weight-loss interventions across all life stages have included advice and strategies to adopt healthy behaviours via diet and/or physical activity. Those adopting an interdisciplinary approach using behavioural strategies and psychological techniques, and specifically targeting the preconception period are almost non-existent. There are also significant barriers to behaviour change regarding healthy lifestyle. Within the target group of women for this study, there is low health literacy, and they are often from low-income families. In a large stratified study on the socioeconomic differences in health behaviours, those with lower socioeconomic status (SES) also displayed less health consciousness (thinking about ways to keep healthy), stronger beliefs about the role of chance on their health and lower rates of thinking about the future [26]. A qualitative study focused on improving health in women of childbearing age identified that dietary knowledge, cooking skills and the time and cost of preparing healthy food were significant barriers to adopting a healthier diet [23]. A lack of support from partners and family members was also cited as a significant barrier to healthy change, likewise, finding the time and inclination to exercise. While preconception care largely focusses on women’s health and care, fathers are increasingly becoming involved in pregnancy planning and lifestyle changes in efforts to conceive. However, little is known about partners’ attitudes and roles in supporting positive preconception health behaviours [27]. A high BMI category is associated with a range of co-morbidities, and poorer general health may affect motivation and perceived ability to adhere to healthy lifestyle recommendations [26]. Behavioural programs have the potential for substantial weight loss, yet significant problems remain due to program attrition and poor maintenance of healthy habits—with authors citing the lack of innovation as a factor in this area [28].

Significant gaps in knowledge exist, with further research exploring women’s perspectives required to inform effective preconception health promotion strategies [29]. This study aims to develop an understanding of preconception health awareness, potential barriers to adopting a healthier lifestyle, motivations, current behaviours and the practical skills required to change behaviour, for women with overweight or obesity. We sought to understand experiences from the women’s perspectives [30] consistent with the aims of the study in fostering empathy with their desires, needs and challenges. The intent is to understand the women’s emotional and social contexts, to inform which behaviour change techniques and intervention components are likely to be most engaging.

## 2. Methods

### 2.1. Theoretical Framework

This qualitative study explored the perceptions and experiences of women related to healthy lifestyle change and weight management before conception. The Information–Motivation–Behavioural Skills (IMB) model [31] addresses some of the existing attitudes, beliefs and values that may impact behaviour [32] and was used as the theoretical model for the interview topics. The model, which asserts that when individuals are well-informed, motivated to act, and hold the necessary behavioural skills, they will likely initiate and maintain a health behaviour [33], is highly applicable to obesity management [32,33]. The IMB model is based on a critical review and integration of relevant constructs in social and health psychology theories and seeks to address limitations to these [33]. The constructs are supported in the literature to improve healthy lifestyle behaviours and have been tested and used successfully in obesity prevention [34,35,36] and in improving dietary and physical activity behaviours [37,38]. This model was chosen for its simplicity of structure and the fact that the constructs can be easily translated into intervention components.

The IMB model’s elicitation–intervention–evaluation approach to the promotion of health behaviour, begins with seeking to identify this cohort’s existing weight management knowledge, motivation and behavioural skills assets and deficits [33]. Individual determinants of behavioural change were explored: Information, including behaviour-related information, knowledge about the impact of obesity, but also heuristics that permit automatic decision-making; Motivation, comprising personal motivation (beliefs about intervention outcome and attitudes towards obesity prevention behaviours) and social motivation (including perceived social support for engaging in that behaviour); and Behavioural Skills (individual skills and self-efficacy) [33].

Qualitative investigation enables a deep and thorough understanding of the topic, yielding rich data [39], and is recommended to develop effective interventions [40]. The relevant literature informed the development of the interview schedule, with probes allowing for exploration of topics driven by participants. Under the belief that knowledge is socially situated [41], demographic information was also gathered, allowing the researchers to reflect on the relationship between the results and the sample [42].

The present research, drawing on the problem-solving principles of design thinking [43], uses a human-centred approach that holds the emotional, functional and motivational needs of the user at the centre of the development process [44]. This study represents the first phase of this process in empathic engagement—with these findings being used to inform the development of an eHealth solution using the intervention mapping approach.

### 2.2. Participants

Interview participants were a sample of women who have participated previously in diet and lifestyle intervention studies (the LIMIT [45] and/or GROW [46] Randomised Trials) at The Women’s and Children’s Hospital—a high-risk specialty hospital with approximately 5000 deliveries annually—and who had given their consent to be contacted about future research. A purposeful sampling frame was adopted, with eligibility limited to those of reproductive age (15–49 years) [47], who were above the healthy weight range (BMI > 25 kg/m^2^) and identified that they would like to lose weight. Intention to become pregnant was not a prerequisite.

### 2.3. Procedure

Women were contacted via telephone, with the purpose and methods of the study explained, and eligibility confirmed. Those who expressed an interest in the research were emailed an information sheet and consent form, with interview times confirmed via email or telephone.

The primary researcher (JS) conducted twenty-three interviews during September 2019. Eight participants attended face-to-face interviews at The University of Adelaide Robinson Research Institute, with the remaining 15 conducted via telephone. The interviews lasted 20–72 min (Mdn = 33). Written consent was obtained from those attending a face-to-face interview, with verbal consent gained from those participating via telephone. The interviews followed after the consent process.

The interviewer was a psychology researcher (JS) trained in qualitative methods and interviewing skills, with knowledge of pre-pregnancy health and wellbeing, and no previous connection with any participants. A pilot interview was conducted with an eligible woman to determine the level of comprehension and natural flow of the intended questions, with this data excluded from the dataset as the interviewee was known to the researcher. After several interviews, some questions were revised for greater understanding, with others modified to broaden the scope for response. No further changes were made after the seventh interview. This process of revision is accepted as best practice within qualitative interviewing [48].

Each interview commenced with broader questions about lifestyle to build rapport. More sensitive and descriptive questions were asked later when participants were more comfortable. The interview schedule closed with questions that empowered the women to give their opinions and advice on the broader issue. Questions were formulated to address the IMB model constructs yet allowed scope for participants to speak freely outside of these topics. Indicative topics included awareness of maternal and neonatal health risks, previous experience of weight loss, lifestyle and social factors, motivations, challenges and self-efficacy. A sample of questions is provided in Table 1.

Participants who attended a face-to-face interview received a 20 AUD gift card as reimbursement for travel costs. After the interview, all women were provided with information on the maternal and neonatal health risks associated with overweight or obesity in pregnancy.

### 2.4. Analytic Approach

All interviews were digitally recorded and transcribed verbatim by the first author. Data analysis software NVivo 12^®^ [49] was used to store and manage the transcripts, with each participant given a pseudonym and identifying information removed from the transcripts. Despite recurring patterns indicating that code saturation was reached after 12 participants, additional women were interviewed to provide meaning saturation—thereby identifying further insights and the nuances of issues required to understand this complex topic [50]. It is believed that depth requires further data, especially for codes that are conceptual in nature [51].

The analysis followed the six-phase process of reflexive thematic analysis set out in Braun and Clarke (2019) [39]: Familiarisation, generating codes, constructing themes, revising themes, defining themes and producing the report. This recursive process begins with repeatedly reading the data in an immersive manner to gain a sense of the whole. The analysis took both a deductive and inductive approach [52]. While guided by the IMB model and obesity literature, the coding process ensured that concepts falling outside of this were also captured. Key concepts relating to the research aims were coded, then sorted into meaningful clusters that were assessed for applicability to the IMB model constructs. Two additional members of the research team (MO, DT) co-coded several transcripts to ensure rigour and transparency of interpretation, with the researchers working collaboratively to ensure the codes and candidate themes fit both the evidence, and the constructs of the IMB model. The themes were then defined and named, with compelling extracts selected to illustrate the findings. Perspectives that differed from the dominant beliefs were not excluded, with counter-instances considered to add rigour [53].

### 2.5. Ethical Considerations and Quality Criteria

The study was conducted in accordance with the Declaration of Helsinki, with the research approved by The Women’s and Children Health Network (HREC/19/WCHN/108) and the University of Adelaide Human Research Ethics Committee. This research was reported as per the consolidated criteria for reporting qualitative studies (COREQ) [54]—a 32-item checklist across three domains governing reflexivity, study design and analysis.

Each author approached the research from their respective positions and biases. While the primary researcher (JS) who conducted the interviews, is not above the healthy weight range, she has an understanding of preconception health. The research team also comprised: a clinical and health psychologist (MO) with extensive experience in preconception care and health literacy; an obstetrician and academic researcher (JD) with extensive experience in maternity care; a digital technology specialist (CS) with an interest in its practical application within the health domain; a clinical trials manager (AD) within reproductive health and preventive medicine; and an academic researcher (DT) with a specialist interest in health psychology and maternity care. All authors are women with children of their own.

A triangulated review process was used, with codes and candidate themes discussed by three researchers (JS, MO, DT). An audit trail was maintained throughout data collection and analysis [41,54], and participant confidentiality ensured by assigning pseudonyms. Participants were able to review their transcripts to verify accuracy [41] and were sent a summary of the findings [54]. Due to the sensitive nature of the topic, and participants not providing consent for sharing of their data beyond the current research, data will not be made publicly-available.

## 3. Results

### 3.1. Participant Characteristics

A total of 23 women, aged between 23 and 48 years (*M* = 34.78, *SD* = 7.33) participated. Each had between one and four children, with two women intending to conceive in the next 12 months and three unsure on pregnancy intentions. Participants self-reported their weight and height—most considered themselves above the healthy weight range (*n* = 22), with BMI categories ranging between overweight (26.1 kg/m^2^) and obese class III (56.3 kg/m^2^). All women expressed a desire to lose weight, with amounts ranging between five and 60 kg. Demographic characteristics of the participants can be seen in Table 2.

### 3.2. Overview

The themes identified in the data reflect the individual perspectives of participants concerning healthy lifestyle changes. These comprise four overarching themes: the IMB model constructs (Information, Motivation, Behavioural Skills) and one issue beyond the scope of these (Environmental Factors) that encompass the diverse experiences of these women. Within these, several themes were derived from the data, depicted in Figure 1. 

Information-related themes included poor health risk knowledge, making healthy food decisions and health versus convenience. Motivation-related themes—divided into personal and social motivation—included concepts such as taking responsibility, flexible options, interpersonal challenges and accountability. Behavioural Skills themes covered concepts such as the mental battle, time management, self-care and inspiration. The women conveyed their feelings and attitudes as they discussed previous experiences, their social environments, and aspects they considered presented barriers to adopting a healthy lifestyle. Themes are presented in greater detail below.

### 3.3. Themes Identified

Information

#### 3.3.1. Theme: Poor Knowledge of Health Risks Impacts Action

Knowledge of the potential maternal and neonatal health risks associated with overweight or obesity before pregnancy was alarmingly low. While many participants had some knowledge of the risks to their health, most commonly gestational diabetes and pre-eclampsia, others expressed this knowledge in more general terms. Of those who knew some of the risks, several disclosed they had experienced these in a previous pregnancy.


*“Um, low birthweight, high chance of gestational diabetes, hypertension. And I’ve got it all, so I’m the statistic [laughs].” (Isabel, Overweight)*


Knowledge of neonatal health risks and outcomes was much less evident, with only half the participants able to recall any at all. Of these, the majority were unable to describe them in detail or they were based on false information. While several women knew the risk of high birthweight, only two participants mentioned that the child may be at risk of having weight issues over their life course. However, some women did express informed knowledge of longer-term health risks for their child.


*“Yeah well, stillbirth, um, high birth weight, um, I might be pulling this out of thin air, but I believe it puts your child at higher risk of having diabetes themselves.” (Amber, Obese class II)*


Even those women who knew some of the risks to their health or their baby’s health were unaware of the genetic traits they may pass on to their child, by entering a pregnancy with overweight or obesity. In many instances, participants only followed a healthy lifestyle after conception—discussing the more immediate effects of the foods they ate during the pregnancy and how this may impact the baby.


*“…you want what’s best for your kids, and that starts from the second you see those two pink lines.” (Annie, Obese class III)*


When asked about the benefits of losing weight before pregnancy, some women expressed that they may fall pregnant more easily, and many conceded the healthier they were, the easier the pregnancy and birth would be, with less strain on their body and fewer complications. Some requested information be freely available on the benefits, for themselves and their baby, of being a healthy weight before conception.

#### 3.3.2. Theme: Making Healthy Food Decisions to Aid Weight Loss

Many of the women had inadequate information on nutrition to make autonomous healthy food decisions for themselves or their families. While no participants were currently on an intensive weight loss program, some foods they reported consuming were carbohydrate-laden or high in sugar or fat—suggesting poor nutritional knowledge and self-monitoring of eating habits. For some, knowledge was based on misinformation such as considering some high sugar or “light” version of foods as healthy.


*“…I suppose an example is when I had gallstones and I wasn’t supposed to be eating fat, I ate kabana [high-fat cured sausage], not realising that it was full of fat, and I ended up in the hospital with a gall bladder attack. It was just a…you know, a lack of information I suppose, I just didn’t know.” (Erin, Overweight)*


Portion control, snacking, willpower and knowledge of consumption norms were among the greatest food challenges—even for those who considered their current diet to be healthy. Positive health knowledge and behaviours were also displayed, with some reporting a good understanding of nutrition to aid weight loss. Several women recognised the value in clean eating and shopping the perimeter of the supermarket to avoid heavily processed foods.


*“Don’t do fad diets, just eat clean. Your food is 80% of it. If you’ve got your food under control, then…other things will fall into place.” (Elizabeth, Overweight)*


While over half of the participants read the ingredients list and/or nutritional information panels, there appeared to be confusion around interpreting these. Many appeared overwhelmed by the volume of nutrition advice available and whether it could be trusted.


*“There’s a whole confusing world out there when it comes to diet” (Mary, Obese class III)*


Several women requested information being more freely available on healthy substitutions for unhealthy ingredients or dishes, plus alternatives to processed foods.

#### 3.3.3. Theme: Managing the Dilemma of Health Versus Convenience

Participants emphasised the complexity of their lives, with busy schedules a contributing factor in choosing convenience meals over healthy ones. Many women spoke of being too exhausted to cook and often relied heavily on processed foods or take-away, while half of the women reported cooking from scratch most of the time. Physical and emotional exhaustion, general busyness and a lack of forward planning were cited as reasons for making poorer choices.


*“…very little time to myself, very little time to prepare food or work out or anything like that. Um, so my lifestyle is very hectic…and very easy… [to choose] what’s easy and convenient, not what is healthy” (Sally, Obese class II)*


Some noted the effects that environmental cues had on human behaviour, particularly when making choices based on ease of preparation and cost.


*“So that reliance on convenience foods, you’re bombarded with it when you walk in the supermarket […] everything that ever goes on special is the cheat food […] Why would I, you know, make my own when I can buy a jar on special for $2.” (Annie, Obese class III)*


Many requested information and recipes to enable them to make fresh, healthy food appealing and delicious—without sacrificing considerable time.

Personal Motivation

#### 3.3.4. Theme: Better Physical and Mental Health for my Family

When asked about their primary motivators for losing weight, some of the women expressed aesthetic motivators over health ones, in wanting to feel good about themselves and more confident.


*“Almost entirely aesthetic. Like I don’t like the way I look in photos. And I understand the health risks, you know, the higher, higher incidence of diabetes, heart disease, all of the fat diseases, but that’s not my main motivator if I’m honest.” (Amber, Obese class II)*


A powerful motivator for nearly all participants was role modelling healthy behaviours to their children, so they can also make good choices. Several women did not want their children to focus on weight or dieting; instead, they wanted to set an example with positive health talk. Within this, a couple of women conceded that the food they fed their children was much healthier than what they ate themselves.


*“What I make my kids eat is healthy [laughs]. What I eat myself is different, I don’t know why, I don’t know why I do it. […] I really would love my girls to see me um, make healthy choices, and live out what I’m making them live [laughs].” (Mary, Obese class III)*


Several women expressed their primary motivation was to improve their overall health, with some conceding that co-morbidities they suffered were an added incentive to lose weight.


*“Just to feel healthier in myself. Also to kick my depression a bit.” (Lucinda, Obese class III)*


Most women cited several reasons for wanting to lose weight—even those with aesthetic motivators also wanted to provide a good example to others and see their children grow up. Other reasons cited were to improve self-esteem, have more energy, feel proud next to their partner or to simply be a normal BMI. Some women looked to the future, in wanting a long healthy life for their children, or more self-sufficient as they age. Several expressed motivators related to the health of their potential baby or having an easier pregnancy.

#### 3.3.5. Theme: Taking Responsibility for My Choices

Most women held positive beliefs about how important it was for them to manage their weight. It was widely recognised that they needed a “lifestyle” rather than a “diet”—having had negative experiences of weight loss programs in the past. Many women attributed their excess weight to unhealthy behavioural patterns and felt a responsibility, whether trying to conceive or not, for managing their weight and health.


*“… trying to conceive, or have already had children and trying to conceive again, you know, you’ve got another body to look after, like it’s not just you any more” (Sasha, Obese class II)*


Many women felt a sense of personal autonomy in choosing to improve their health—recognising that their lifestyle choices were modifiable and to have a healthy life, they had to take stock of their habits and change their mindset.


*“I don’t blame my kids or my husband and I don’t think that I don’t have enough time.” (Mary, Obese class III)*


Many were ready to change their lifestyle but perceived a barrier in making that first step—not knowing where to start in a task that seemed overwhelming. Beliefs about other parts of women’s lives often affected their motivation to begin a program of change.


*“Again, it’s just that overwhelming feeling of, everything’s wrong and I can’t fix any of it, where do I start, you know? […] I’m defeated in my body, I’m defeated in my mind, I’m defeated in my, you know, um, in my home. […] I can’t get on top of anything, and so I will start nowhere [laughs].” (Mary, Obese class III)*


Some women conceded they had no reason to eat unhealthily and were ignoring satiety cues or lacking in motivation.

#### 3.3.6. Theme: Simple, Flexible Options Enhance Motivation

All participants had previous experience with trying to lose weight. These included meal replacement shakes, the ketogenic (or other low carbohydrate) diet, Duromine or other diet pills, the “Healthy Mummy” app, intermittent fasting, Weight Watchers and counting calories. Most had tried at least two of these methods. Although some women had experienced success, many failed to lose weight or found the programs too restrictive, costly, or rigid to maintain with other responsibilities. Women required options for both healthy eating and exercise that were flexible enough to accommodate family commitments.


*“I like taking the ideas from it, but I need the flexibility, with the family…for what’s best to eat for us.” (Chloe, Overweight)*


Women talked about establishing a new routine that became an automatic habit. Several women noted that they wanted a program that they could use long term to manage their weight.


*“I don’t want a diet that I can’t have for the rest of my life.” (Mary, Obese class III)*


Some noted that small steps and achievable goals would make women more motivated to succeed, along with devising a program that was simple to follow.

Social Motivation

#### 3.3.7. Theme: Feeling the Pressure of Social Expectation

Many women reported feeling that societal norms dictated how they should look and feel and felt judged when they did not fit the “social” mould. Some felt they were perceived differently due to their weight and did not want to shame their children or partners.


*“You’re bombarded with Weight Watchers, Michelle Bridges, the Biggest Loser, um…all of that, social media bombards you every day. […] Yeah, and that societal norm. […] there’s always going to be that voice in the back of your head that there is someone judging you, that doesn’t even know you.” (Annie, Obese class III)*


A handful of participants could not relate to some of the advice or services they had accessed in the past, believing they target women who were already fit and a healthy weight, and found this de-motivating. Despite experiencing hostility from others, or feeling out of place in fitness classes, some women were still able to push forward to improve their health.

#### 3.3.8. Theme: Interpersonal Challenges Can Affect Motivation

Several women noted their partners were a barrier to eating more healthily, despite how supportive they may be in other ways. Either they prepared unhealthy foods or ate unhealthy food in front of them.


*“I struggle every day to eat well [laughs]. […] My partner is 60 kilos wringing wet and can sit there and eat whatever he likes. And it’s like trying to give up smoking in a house…where people smoke. […] he’s never been overweight so he doesn’t understand.” (Erin, Overweight)*


More than half of the women reported that their children were fussy eaters—which provided an added barrier to motivation, in that mothers preferred to cook food they knew their children would eat, at the expense of their own health. Some struggled to strike a balance between healthy foods and the nutritional needs and preferences of their children, so as not to have to cook separate meals.


*“I’ve got two kids that are fussy eaters, to try and provide foods that are easy and carb-loaded for them […] So that’s where I‘ve had issues with putting on weight, because trying to motivate the kids to eat and put on a bit of weight, you have to eat what they’re eating. So, it’s just trying to get that balance.” (Ella, Obese class II)*


Some had difficulty convincing others of the need for healthy eating and felt this attitude sabotaged any efforts to lose weight.

#### 3.3.9. Theme: Encouragement and Accountability Keep Motivation High

Many women reported ongoing encouragement as holding power to maintain motivation—regardless of whether they saw results—citing it as a significant reason not to abandon a healthy lifestyle program.


*“Just nice words of encouragement, like you know ‘you’re doing a good job, and well done on you know, getting through the day, and you know, reaching your goal of however many steps’ or ‘it’s ok you didn’t get there today, but, you know, you’re still doing good’.” (Olivia, Overweight)*


Women often find motivation wanes when “life gets in the way” and wanted gentle encouragement to keep on track after a setback. Methods were highly individualized—while some wanted strategies that challenge on an individual level with personal messages, others found a group setting or competitive environment more motivating in that they saw others push themselves. The most powerful motivational support for women was seeing results—several had abandoned previous weight loss attempts due to not seeing results quickly enough. Overwhelmingly, women felt they needed to be held accountable—some preferred a real person rather than goals set in an app, feeling they may be more likely to move the goalposts or lose motivation if they could not reach them.


*“I can fool myself pretty easily and make excuses for why I haven’t done what I’ve done. It’s different when you’ve got to explain to someone else your pathetic reasons for not having done something.” (Amber, Obese class II)*


Several participants saw value in family members also committing to improving their health habits, to create an environment more conducive to success. Some saw reciprocal benefits in encouraging others—such as peer interaction within an online support forum—and valued the camaraderie with those on the same journey.

Behavioural Skills

#### 3.3.10. Theme: Overcoming the Mental Battle

Unprompted, several women mentioned psychological support in managing setbacks, stress and maintaining motivation—knowing the difficulties that can be encountered.


*“…mental health is a huge thing when you’re trying to lose weight. […] [it] can be really demoralising, especially if you’re not achieving those goals. Or if people around you are achieving them, and you’re not.” (Elizabeth, Overweight)*


Weight management was viewed as much more complicated than just calorie input versus output. One participant noted that for most, obesity is about more than just food, and she would value being able to talk to someone about other issues that may impact behaviours. Some women talked about their mental barriers and expressed the need to change their thinking around old habits.


*“You’ve really got to sort of train yourself mentally as well, to…um, eat better and to exercise more. It’s kind of a mental battle as well.” (Olivia, Overweight)*


Participants overwhelmingly had stressful lives, with factors such as juggling work and caring for children, financial difficulties, children with complex health issues, or managing a hectic schedule contributing to this. For some, exercise was noted as being particularly beneficial for stress, with several women requesting stress management or mindfulness approaches within an intervention. Some also valued a holistic approach incorporating increasing self-esteem too, with the belief it would help maintain good health decisions.

#### 3.3.11. Theme: It Would Be Easy, with Better Time Management

Most women recognised time as one of the biggest barriers to a healthier lifestyle. Prioritising their health over other responsibilities proved difficult and contributed to the abandonment of previous weight loss attempts. Some participants recognised that exercising would be easy if they were able to manage their time better.


*“I feel like it’s just managing my time better isn’t it? Just getting up, even like a little bit earlier, I can do it in the morning…” (Alex, Obese class II)*


Some felt that time spent cooking from scratch or exercising meant that other priorities in their lives piled up and left them further behind. Several women conceded that they do not really have an excuse not to eat well or exercise, but probably do not make the best use of their time, with “perceived busyness” often an excuse.


*“I think if someone can help you plan your whole day, so you can fit it in.” (Mary, Obese class III)*


Exercise proved more of a challenge than healthy eating, and women wanted ideas for how to integrate physical activity into their lifestyle with small changes. Women often felt a trade-off between spending time with their children and exercising, with the unpredictable nature of family life a barrier to maintaining physical activity and healthy eating regimes.

#### 3.3.12. Theme: Taking Care of Myself Is Important as I Value My Health

Many women also recognised other benefits, beyond weight management, in eating well and exercising. Participants noted these self-care behaviours were rewarded with improvements in sleep, energy, clearer skin, mental clarity, less digestive issues, and some felt it gave them more motivation in other areas of their lives. Overwhelmingly, women assigned greater importance to eating well than physical activity.


*“When I eat well, I feel well if that makes sense.” (Erin, Overweight)*


Some noted it took longer to notice the health benefits of physical activity, yet several women placed higher importance on exercise than healthy food, for the additional psychological benefits.


*“For me it’s got the stress relief component, it’s my ‘me’ time, it feels good, I feel good afterwards, I tend to be…I find the flow on effects, so when I exercise that’s when I do tend to be less likely to go to straight the cupboard because I’m still feeling motivated.” (Chloe, Overweight)*


It was also mentioned that, as mothers, they often prioritised everything else before their health. Some found value in exercise as it allows them to focus on themselves, rather than just taking care of their families.

#### 3.3.13. Theme: I Need More Inspiration Than Information

Many women lacked skills such as nutrition planning and meal preparation. Participants indicated they often found healthy food boring or repetitive and wanted ideas to make meals more interesting and attractive to themselves and their families.


*“…knowing how to make foods interesting so you’re not eating the same old things continuously […] how to make vegetables more interesting […] without adding massive amounts of calories to them.” (Lucinda, Obese class III)*


Around half the women regularly cooked meals from scratch with fresh produce. While not always intentional with their meal planning, a handful of women noted they were skilled in creating basic meals, based on ingredients at hand and their personal preferences. Some participants noted that what they needed was inspiration to implement changes.


*“…oh gosh, I need more inspiration than information” (Amber, Obese class II)*


Several women expressed a desire for basic meal plans and simple, quick recipes that were within their skillset, using easily accessible, cost-effective ingredients. Women knew their old, unhealthy patterns of eating, and highly restricted regimes were not sustainable in helping them establish healthy patterns for weight loss. Many wanted a more structured plan with interesting ideas for both nutrition and exercise.

#### 3.3.14. Theme: Believing in Myself to Change My Lifestyle

Several women noted they felt confident to make healthy changes to their diet and exercise levels, but motivation held them back. Many placed caveats on their ability—contingent upon managing their time better, managing exhaustion, or finding adequate motivation.


*“Very [confident]. As long as it’s cost-effective and easy to fit it in with the schedule. Like it’s not a 2 h prep. […] It’s gotta be a…let’s just chop it up and get it happening. But I like things from scratch.” (April, Overweight)*


For some, previous successful experience of weight loss gave them confidence that they could make the changes required.


*“Reasonably confident. I know I can do it. I know I can, because I’ve done that with little things, like the soft drink as I said. So I know I could do it.” (Erin, Overweight)*


Women often reported better results with changing their diet than exercising and expressed more confidence in being able to modify their diet. Several cited the 80/20 rule—believing that weight loss is 80% diet and 20% exercise. Given the correct information, many felt confident about easily integrating healthy foods into their daily life. However, a handful of women recognised that their lack of cooking skills might hinder their progress. Some felt it would be easy to include physical exercise in their daily life, but only after attending to other responsibilities.

Environmental Factors

#### 3.3.15. Theme: Affordability Limits Access to Healthy Options

Several women considered healthy food to be more expensive than convenience foods, and some cited this as a reason for their poor diet at times. They noted that the foods on special offer were the “cheat” foods, not the healthier options, and this influenced their purchase choices.


*“So it’s more about finding the alternate to those expensive things […] because it’s very expensive starting a diet and that can sometimes put a lot of people off, because budget-wise you just can’t fit it in. And I find that’s why people go for the easy foods, because it is cheaper.” (April, Overweight)*


Especially for families with limited income, cost encouraged reliance on cheaper fat-, sugar- or carbohydrate-laden foods. Frozen or packaged food was sometimes bought in favour of fresh, as it could be relied upon if money was tight. Leaner cuts of meat were also seen as less affordable.


*“…that makes it a bit harder sometimes, especially if we’re like, tight on money and the only thing that we can whip up is like sausages and packet pasta and stuff like that.” (Carla, Obese class III)*


Some were unable to continue with previous weight management programs due to food costs. Physical exercise often presented a cost barrier too, with several noting the cost of gym memberships, boot camp and exercise clothing suitable for overweight women.

## 4. Discussion

This qualitative investigation examined the experiences and beliefs of women with overweight or obesity related to managing their weight before pregnancy. In general, the participants displayed poor health literacy on the impact of entering pregnancy with overweight or obesity. While some women were aware of the risks to their health, few recognised the potential risks to their future baby. This finding corresponds with previous research noting a poor understanding of neonatal outcomes for pregnant women with overweight or obesity [55]. While preconception care and counselling may include advice about smoking, alcohol intake, nutritional supplementation and immunisations, less attention has been paid to diet and lifestyle advice for women with overweight or obesity [56]. The topic holds such sensitivity that it is often not raised by doctors during consultations.

Previous systematic review research has found that the beliefs and attitudes of partners, peers and family exert a powerful influence on women’s health behaviours that may undermine the advice of health professionals [57]. The fact that many women reported interpersonal challenges suggests that partners need greater involvement in the process of preconception counselling and creating and supporting a healthy lifestyle. Pregnancy is considered a powerful “teachable moment” [58] for weight control and health behaviour change, which enhances the perceived value of nutrition and exercise. While the focus of the current study has been on efforts women can undertake in the preconception period to maximise outcomes for their child, pregnancy is often a joint endeavour and partners are becoming increasingly involved in preconception planning. It is evident that partners can help or hinder women in their efforts to make lifestyle changes. It is crucial that any future intervention consider the impact of partners including ways to foster social support by making lifestyle or behavioural changes themselves, and to reduce social pressure by lifting the stigma around weight. This would provide a supportive environment to initiate healthy change within the whole family.

Interestingly, many of the women conceded they were motivated more by aesthetics and social expectation than health considerations. This finding reiterates previous research that found motivations beyond health, such as self-image, may be more engaging for some women [59], and enable those not yet considering pregnancy to be captured by the public health messages. While it has been reported that between 56% [60] and 70% [29] of pregnancies are planned, it is thought that women with overweight or obesity are more likely to have unplanned pregnancies [61] and therefore are less able to optimise their health before conception. Thus, interventions may need to target not just those intending to conceive, but all women of childbearing age. For many women in the study, the prime consideration was being healthy during the pregnancy, rather than in preparation for conception. Greater understanding of the health implications of weight status may provide an added motivation to improve lifestyle.

While the current study is concerned with behaviour change on an individual level, interactions with surrounding influences—family, community, plus social, environmental and policy contexts—exert a powerful force on health behaviours, as noted by many of the women in this study and previous research [12]. A two-pronged approach is required—empowering women with the tools to change the way they respond to the environment, and also changing their environment, where possible, to make healthier choices easier to make. The sugar tax, intended to create shifts in consumer behaviour, has reduced sugar consumption in the United Kingdom and Mexico [62], with higher effects for lower-income households. While food taxes and subsidies have not yet been implemented in Australia [63], it stresses the importance of giving women evidence-informed guidance they can trust, but also individual support to counteract obesogenic environments [64].

Unsurprisingly, nutrition knowledge was relatively poor amongst participants, with many consuming processed foods high in sugar or fat. It is known that unconscious and instinctual processes can prompt poorer eating practices, with people relying on heuristic cues to make food decisions which often lead them to choose larger and less nutritious options [65]. Heuristics are mental shortcuts—in this instance, made in response to contextual food cues and are thought to reduce the cognitive depletion associated with making health decisions [65]—especially salient in a population with lower health literacy. Many women had requested simple solutions, rather than complex calorie-counting and food logging. These findings emphasise the need for interventions to respond to the preference for heuristic processing, to help people make better choices with regard to food, portion control, even leisure activities, as suggested in a previous study [65].

Previous experience of dieting practices meant that many of the women were wary of highly restrictive eating regimes, which often affected adherence and success. This result corresponds with previous research on barriers to following a Mediterranean style diet for women of childbearing age [23], with women perceiving the term “diet” to have negative connotations. The language used will be critical to the success of any future intervention, a sentiment shared by previous research on public perceptions of the terms used to improve eating habits [66]. This information highlights the need to frame the intervention with clear communication and a positive focus, along with making guidelines more flexible to accommodate busy lifestyles and families, as noted by previous research [67]. Accessibility was also problematic, with the perception that healthy food was more expensive—a finding common to other studies [23,29]. This finding points to the need for education on better food choices and substitutions—being mindful of budgetary constraints.

Women had stressed the importance of social support, encouragement and accountability, suggesting the need for an inclusive community to be created, where women feel empowered and supported to reach their goals—as noted in previous research on mutual-help groups [68]. Avenues for peer support have never before been so important, with social barriers such as those imposed by the COVID-19 pandemic, along with caring responsibilities for other children, giving women less opportunity to interact with peers and health professionals for encouragement.

Many participants also recognised the benefits of a healthier lifestyle beyond weight loss itself. A holistic intervention therefore presents an opportunity to promote factors such as stress reduction, increased self-esteem, improved mental health or role modelling. Several women in the study expressed a need for psychological support along their weight loss journey. A technology-delivered motivational interviewing approach [69] could provide tactics to increase self-efficacy through small steps and achievable goals, elicit change talk and garner family support. A regime of establishing and recognising partial success in changing behaviour, plus strategies to manage setbacks, may promote greater motivation to continue. Mindfulness-based interventions may have particular relevance for obesity with a meta-analysis suggesting benefit for improving both psychological health and eating behaviours [70]. Likewise, open trial and case series evidence suggests that Cognitive Behavioural Therapy (CBT) [71] and Acceptance and Commitment Therapy (ACT) [28] can provide useful adjuncts to other behavioural techniques for weight management. However, few studies exist that assess these techniques when delivered via digital health intervention.

An eHealth intervention offers a solution that is low cost, high reach, with the potential for personalisation and the use of adaptive and agile design to improve efficacy. This study presents an opportunity to understand the behaviour change techniques and digital functionality to which women may best respond. In-depth qualitative research is crucial in understanding the personal experiences and the contexts within which potential intervention users live, and to help tailor interventions to specific life stages. This person-based approach complements the theory- and evidence-based approaches to intervention development [72]. An intervention mapping approach will be guided by these interviews, the literature on obesity, behaviour change and psychological techniques. It is hoped that themes and subthemes derived from this study, having been identified by the women as important, will directly inform modules to be delivered in the intervention. The intent is to develop an intervention that the women want to engage with, using strategies that address some of the existing barriers to change and help them to create a sustainable healthy lifestyle.

### Strengths and Limitations

To date, there has been little qualitative research conducted into the experiences and beliefs of women regarding weight management before pregnancy—within an Australian context. To the authors knowledge, there have been none that use the IMB determinants of behaviour within this specific target group.

This cohort displayed a diverse profile across age, family circumstance, BMI and socioeconomic factors. Credibility was added through the multivocality of the participants, with the authors aware of the empathic understanding required to let this wide range of insights emerge. Both the interviews and analysis were conducted with rigour, adhering to best-practice guidelines for qualitative research [41,54].

The research was not without limitations. The women interviewed were not necessarily intending pregnancy (*n* = 2 intending pregnancy in next 12 months, *n* = 3 unsure, *n* = 18 not planning pregnancy in the next 12 months). Therefore, motivations may be different from those in the preconception phase. However, it is thought that similar health and role-modelling sentiments would stand for those intending pregnancy, in wanting the best outcomes for their children.

The women were recruited from a pool who had previously participated in studies concerning gestational weight gain, so may have prior knowledge of managing their weight through pregnancy in a supported manner. In addition, some women reflected on previous pregnancies in their responses, discussing lived experience of the health risks associated with their weight status. Moreover, the cohort was not culturally diverse, in that recruitment was limited to those who spoke English. The cultural implications of dietary and social habits need to be considered and, therefore, future research with diverse populations is recommended.

## 5. Conclusions

The preconception period is now acknowledged as a critical window in which to intervene for preventing obesity in pregnancy, with significant potential benefits—both health and economic. However, obesity is a complex and challenging issue, with multiple genetic, social, environmental and behavioural influences. Promoting meaningful change in this group requires a multifactorial approach, involving a complex interaction between the necessary determinants of behaviour—information, motivation, and behavioural skills. This elicitation study, the first step in the IMB approach to health behaviour change, provided insights about the beliefs and psychosocial contexts of women in this particular population. Important factors for consideration include psychological support, flexibility, enhancing self-efficacy, motivational support and affordability. A tailored, empathic and collaborative intervention approach will be taken, guided by the perspectives gained from this study. Informed by the current findings and the existing literature, we seek to develop an effective digital health intervention, that results in improved multigenerational health outcomes for women and their children.

## Figures and Tables

**Figure 1 jcm-09-03351-f001:**
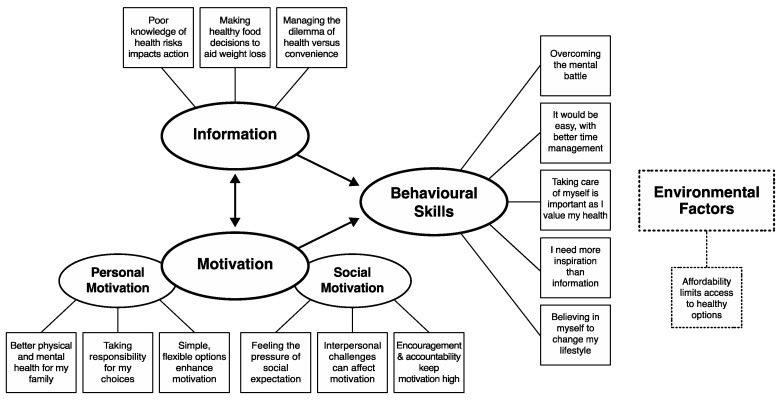
Thematic map.

**Table 1 jcm-09-03351-t001:** Indicative semi-structured interview questions.

Tell me a little about your lifestyle at the moment. What do you know about the potential risks to your own health, of being overweight going into a pregnancy? What are your reasons for wanting to lose weight? Do you consider your current diet to be healthy? What other information do you believe may support you to develop a healthy eating plan? Are there ever times when you feel it is more difficult to eat well and exercise? How confident do you feel about being able to exercise more? If you could give one piece of dietary advice to another woman who was considering losing weight before pregnancy, what would it be?

**Table 2 jcm-09-03351-t002:** Demographic characteristics of participants (*n* = 23).

	*n*
Age (years)	
20–29	7
30–39	9
40–49	7
Number of children	
1	4
2	8
3	6
4	5
Intending pregnancy in next 12 months	
Yes	2
No	18
Unsure	3
Body Mass Index (BMI) ^a^	
Overweight (25.0–29.9) ^b^	9
Obese class I (30.0–34.9) ^b^	4
Obese class II (35.0–39.9) ^b^	5
Obese class III (≥40.0) ^b^	5
Amount of weight desire to lose (kg)	
5–9	5
10–19	6
20–29	2
30–39	4
40–49	2
50+	3
Unsure	1
Education	
High School	4
Trade or Diploma	16
Degree	2
Postgraduate Degree	1
Employment Status	
Full time	6
Part time	6
Student	2
Home duties	9
Family situation	
Single parent family	1
Two-parent family	21
Undisclosed	1
Difficulty meeting basic living costs	
Weekly	4
Monthly	8
Never	11

^a^ Body Mass Index (BMI), calculated as: weight (kg)/(height (m))^2^, ^b^ BMI Categories: Overweight: BMI 25.0–29.9 kg/m^2^; Obese class I: BMI 30.0–34.9 kg/m^2^; Obese class II: BMI 35.0–39.9 kg/m^2^; Obese class III: BMI ≥ 40.0 kg/m^2^.

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
