# Peer review of "Creating Healthy Change in the Preconception Period for Women with Overweight or Obesity: A Qualitative Study Using the Information–Motivation–Behavioural Skills Model"

_jcm, 2020, doi:10.3390/jcm9103351_

Round 1

Reviewer 1 Report

I have had the chance to review your manuscript, which I read with interest.  On review, I have the following comments:

GENERAL:

  • Please try and avoid the terminology of 'emerging' for your themes.  Content analysis is not that conceptual, working on the basis of presence or absence of a term or concept, therefore 'emerging' is misleading.

ABSTRACT:

  • Please qualify your opening statement: half of women, where?  Globally?  Just in Australia?  I assumed it to be globally, as this is how you start your introduction - but it is not clear.

INTRODUCTION:

  • Very clear and well written.  Sets up the rest of the paper nicely.  Informative, yet concise to holds the reader's attention.

METHODS:

  • I find the use of the IBM model interesting, but I'm not sure Content Analysis will do it justice.
  • You state that you are using qualitative investigation to achieve 'rich data' and to 'develop interventions' (page 3; lines 125-126), yet you have employed the least complex (and least philosophically 'qualitative') qualitative methodology - content analysis.
  • Your definition of reproductive age being 18-50years is arbitrary and will need re-phrasing or a citation for why you chose this age range.
  • Page 4, Line 149 is a really clunky way to phrase the sentence, please re-phrase.
  • NVivo is stylised as so, not as you have written it (page 4, line 176).  Also, what version of NVivo.
  • Please explain how and why you used inductive and deductive approaches - because it is not best practice to do so.
  • Why did you settle on 23 participants - there seems to be no rationale, and one could argue that for a simple content analysis, it is not actually enough participants.
  • You cite Hsieh & Shannon (2005) - but you do not explain which content analysis you are using from their paper.  They discuss three approaches, all of which are very different and so you do need to explain which you followed and the differences between them which led to you not choosing the other two.
  • Your section on Ethical considerations and quality criteria, is verbose and could do with truncating, substantially.

RESULTS:

  • I think your themes are good - but some of your sub-themes are questionable, as some are so specific, I struggle to believe that they were common across the dataset (e.g. I deserve to feel good).  I would delete your sub-themes altogether and present solely themes.  You only present themes, so I am unsure of why you have even mentioned sub-themes.
  • Your results appear to read as if you present the quotation and do the explanatory text after it, which gives your results section a bit of a stilted read.  Usual practice is that there is a statement made by the author which is then supplemented by a quotation (after).  the next piece of writing would link the quotation to the next statement made and the next quotation.  This does not happen all the time in your paper, and I find myself reading quotations which have nothing to do with the text above them, but linking to the text below them, which is an unusual - and I can't say an easy - way of reading qualitative work.
  • I think you have made quite a jump assuming that women understood the 'health heuristics' associated with their habits and their health condition (page 9, line 290).  There is no evidence of this in the quotation supporting this statement - the participant does not make this link and neither should you.  Don't overreach your findings.
  • Your results section is incredibly long, running from page 5 to page 14.  This is out of step with content analyses, which often report numbers associated with themes and a reduced display of quotations. 
  • Sometimes you end on a quotation, sometimes you end on summary text.  Do one or the other, but not both, it looks inconsistent and messy.
  • You must present the number of women (n=XX) in which the themes were found.  You have done this on some (page 9, line 305), but for content analysis the reader needs to know the numbers.  If you wanted it to be more conceptual, you should not have used content analysis.  I actually think you have done a pretty good jon of the analysis, but I would like to see the representative numbers.

DISCUSSION:

  • Generally, good - nice linking into existing literature and positions your work well.
  • Could be a bit more concisely written, as is quite lengthy for a discussion.
  • I'd like to see a bit more reflection on the utility of qualitative work on designing interventions and how it produces the foundation on which to do that.  How can future qualitative work inform the intervention or the development processes.

CONCLUSIONS:

  • Great, concise ending to your paper.  This level of conciseness should be replicated throughout the paper, in sections which I have identified above as overly long.

REFERENCES:

  • You have quite a lot of old references - 20% (13) published before 2010 and 32% (21) published between 2010 and 2014; meaning that 52% of your references are older than 5 years, which on reflection gives your paper a dated feel.  Please refresh older references where appropriate and possible.

Reviewer 2 Report

Dear authors, 

It is a well-written article adressing a scientifically and clinically relevant topic. 

I formulated some minor suggestions:

  • line 39: the semicolon should be a right bracket.
  • The introduction could have a bigger emphasis on the importance of the preconception period as the 'window of opportunity'. Explain something about the theoretical background; gamete development, epigenetic programming. Then it will become more clear why the preconception period should be the time to target an unhealthy lifestyle. 
  • In the introduction (and the study itself), no attention is paid to the partner/future father and his (supporting) role in making lifestyle changes. In the discussion, you mention the influence of partners on women's health behaviors. However, did none of the participants mentioned during the interviews what they needed or expected from their partners to enhance lifestyle change?
  • line 225: is the weight of study participants measured by a healthcare professional or asked during the interviews?
  • line 267: what do you mean with genetic risks? what kind of answer did you want to hear from your participants?
  • Discussion: did you consider to mention something about psychological therapies to change lifestyle behavior, such as cognitive behavioral therapy, motivational interviewing or minfulness?

Reviewer 3 Report

I appreciate this interesting work, and find it overall an interesting and well-designed study. You addressed the lack of planning of an upcoming pregnancy for a large number of the participants, which as you note could make a difference in how they view the issue of overweight/obesity, and thus kept responses specifically about pregnancy-related weight issues fairly minimal, although certainly present. I found myself thinking throughout your findings, "This is really applicable to women's lives in general."

Intro - Nicely written, very clear but could benefit from subheadings to guide your reader a little more directly through this long section. 

Framework - It is appropriate to your work; however, a figure showing the concepts and linkages would be helpful. Recognizing that you needed to add "environmental factors" was significant and gave greater depth to your findings. 

Aim statement - Very clear. 

Sampling strategy - I understand what you did in randomly selecting participants from your previous study, but this understanding required a second look once you mention purposeful (a less awkward word that "purposive") sampling. The confusion lies in setting your readers' expectations via the use of "random"  = quantitative design, then proceeding to describe your qual technique. 

Findings - Good use of quotes as exemplars supporting your findings.

3.3.7, last sentence re: resilience and strength. This read more like your interpretation rather than data. 

3.3.13. inspiration v. information - this is the least convincing of your themes. I am not sure that the quotes you use as examples support this theme. 

Overall, this paper is highly readable; however, if the journal allows first person singular or plural, your writing would be smoother are clearer. The heavy use of passive voice and anthropomorphisms are distracting and interrupt the narrative flow; however a good editor could help you remedy this fairly easily. 

Round 2

Reviewer 1 Report

Thankyou for working so diligently to make the required revisions from myself and the other two reviewers.  Great work!

I have had the time to read your responses to my comments and to those of the other reviewers and am content that they have sufficiently improved the paper to the point where I am happy to endorse for publication.

I have two remaining revisions - incredibly minor:

Pages 5-6 of the revised manuscript (lines 224-232), please revise sentences so that names are presented as initials only in parentheses rather than full written names.  i.e. "The primary researcher (JS) who conducted the interviews..." and so on.

Please remove (n=) on page 8, line 285 & 292-293;  page 9, line 306

Author Response

Dear Reviewer

Many thanks for your comments and revisions. The minor revisions noted below have been updated as enclosed.

The authorship team all agree the manuscript has been greatly improved by the reviewer comments and suggestions.

Regards

Jodie